

# Field-induced magnetic states in geometrically frustrated SrEr$_2$O$_4$

N. Qureshi[1], O. Fabelo[1], P. Manuel[2], D. D. Khalyavin[2], E. Lhotel[3],
S.X.M. Riberolles[1,4], G. Balakrishnan[4] and O. A. Petrenko[4*]

**1** Institut Laue-Langevin, 71 Avenue des Martyrs, CS 20156,
38042 Grenoble Cedex 9, France
**2** ISIS Facility, STFC Rutherford Appleton Laboratory, Chilton,
Didcot, OX11 0QX, United Kingdom
**3** Institut Néel CNRS, Université Grenoble Alpes, 38042 Grenoble, France
**4** Department of Physics, University of Warwick, Coventry CV4 7AL, United Kingdom

⋆ O.Petrenko@warwick.ac.uk

## Abstract

We report an unusual in-field behaviour of SrEr$_2$O$_4$ for a magnetic field applied along two high-symmetry directions, the *a* and *c* axes. This geometrically frustrated magnet hosts two crystallographically inequivalent Er ions, Er1 and Er2, that are both located on triangular *zigzag* ladders, but only one site, Er1, forms a long-range magnetic order at low temperatures in a zero field. We follow the sequence of peculiar field induced states in SrEr$_2$O$_4$ with detailed single-crystal magnetisation and neutron diffraction experiments. On application of an external field along the *c* axis, the long-range antiferromagnetic order of the Er1 ions is rapidly destroyed and replaced, in fields between 2 and 5 kOe, by a state with shorter-range correlations. The change in correlation length coincides with a fast increase in magnetisation during the metamagnetic transition above which a long-range order is reestablished and maintained into the high fields. The high-field ferromagnet-like order is characterised by significantly different magnetic moments on the two Er sites, with the Er1 site dominating the magnetisation process. For the field applied parallel to the *a* axis, in the field range of 4 to 12 kOe, the planes of diffuse magnetic scattering observed in zero field due to the one-dimensional correlations between the Er2 moments are replaced by much more localised but still diffuse features corresponding to the establishment of an *up-up-down* structure associated with a one-third magnetisation plateau. Above 14 kOe, a ferromagnet-like high-field order is induced following another phase transition. For this direction of the field, the Er2 moments dictate the succession of transitions while the Er1 moments remain significantly less polarised. A complete field polarisation of both Er sites is not achieved even at 50 kOe for either field direction, reflecting the strongly anisotropic nature of magnetisation process in SrEr$_2$O$_4$.

## 1 Introduction

It is not unusual for magnetic systems to demonstrate contrasting behaviour in magnetic fields applied along different crystallographic directions. In fact, considerable anisotropy of magnetisation is present in most magnets, however, it is rather unusual for a magnetic system to show completely different physics governing the magnetisation process for the two field directions. We report the results of the investigation of the magnetic properties of $SrEr_2O_4$ which provides a rare opportunity of observing two contrasting phenomena for the two orthogonal field directions, a metamagnetic transition corresponding to a spin-flip for one direction and a stabilisation of the up-up-down (UUD) structure, usually found in triangular antiferromagnets, for another. Most peculiarly the field-induced UUD state formed by one half of the magnetic Er ions appears to be embedded in an ordered antiferromagnetic state formed by the remaining half.

$SrEr_2O_4$ belongs to the large family of rare-earth (RE) strontium oxides, $SrRE_2O_4$. Interest in the low-temperature magnetic behaviour of this family was initially been sparked by the paper of Karunadasa *et al.* [1]. Later studies revealed a wide variety of magnetic behaviours depending on the choice of the RE ion, ranging from the absence of long-range order in $SrDy_2O_4$ [2–5], the coexistence of two types of short-range order in $SrHo_2O_4$ [3,6,7], and a noncollinear antiferromagneticic structure in $SrYb_2O_4$ [8]. The most interesting results for $SrDy_2O_4$ have been found in an applied magnetic field, where the long-range order is stabilised out of a short-ranged zero-field state [2,9–11]. A somewhat similar field-induced magnetisation process (and perhaps the one closest to $SrEr_2O_4$ in character) has recently been reported for $SrHo_2O_4$ [12], although the local environments are significantly different for the RE magnetic ions in these two compounds.

The key ingredient for the unconventional behaviour of the $SrRE_2O_4$ magnets is the presence of two slightly different crystallographic environments for the RE ions in the orthorhom-

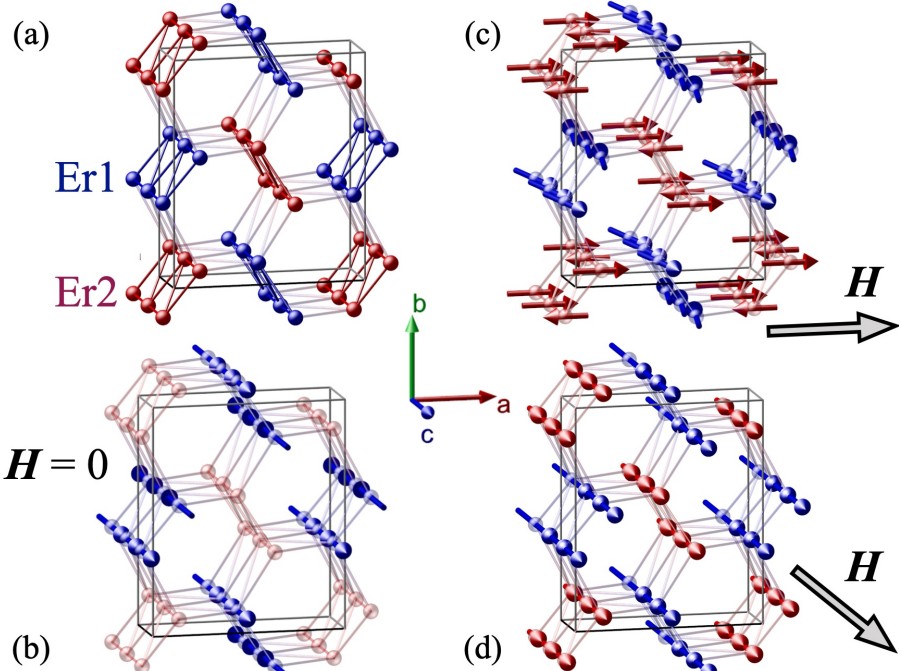

Figure 1: (a) Geometry of the *zigzag* ladders formed by the Er1 (blue) and Er2 (red) magnetic ions in SrEr$_2$O$_4$. (b) Long-range antiferromagnetic structure formed by the Er1 ions in zero field where the Er2 ions remain largely disordered (although they do participate in a formation of weakly correlated spin-chains). (c) Field induced UUD order of the Er2 ions emerging in almost undistorted zero-field structure of the Er1 ions for the field between 4 and 12 kOe applied along the *a* axis. (d) Field polarised structure for $H \parallel c$ in which both Er1 and Er2 ions carry significant but still different magnetic moments.

bic unit cell (space group $Pnam$). The effects of crystal fields are such that the directions of magnetic easy axes (or easy planes) are often orthogonal for the two sites [3, 13]. In SrEr$_2$O$_4$ in particular, relatively small variations in the positions of six nearest-neighbour oxygen ions which form distorted octahedra around the Er ions on two sites results in the occurrence of a strong Ising-like anisotropy with the easy-magnetisation direction along the *c* axis at the Er1 sites and along the *a* axis at the Er2 sites [13]. The RE ions on each site form triangular *zigzag* ladders running along the *c* axis, while the distances between the ladders (bonds between the ions on different sites) are slightly longer, therefore making the interactions between them weaker. Linked together, the *zigzag* ladders form a distorted honeycomb structure when viewed in the *a*-*b* plane, see Fig. 1a.

In SrEr$_2$O$_4$, the magnetic moments on the Er1 sites prefer to point along the *c* axis [13], and then below $T_N = 0.75$ K, they form a fully ordered antiferromagnetic state characterised by a $\mathbf{q} = 0$ propagation vector as shown in Fig. 1b [14, 15]. The magnetic moments on the Er2 site prefer to point along the *a* axis [13] and participate in the formation of a short-range incommensurate structure, which consists of a collection of weakly-correlated antiferromagnetic chains [15]. The incommensurate chains are nearly antiferromagnetic, characterised by the $\mathbf{q} = (0\ 0\ \frac{1}{2} + \delta)$ propagation vector with $\delta$ always small and varying between 0.02 and 0.06 with temperature [15]. We show below that for the two cases considered in this paper, $H \parallel a$ and $H \parallel c$, the magnetisation process is mostly governed by the rearrangements of the magnetic moments on the Er2 and Er1 sites, respectively, while the contributions to the magnetisation from the other site are less significant. It is rare to find such a clear illustrations of

two completely different fundamental processes, one associated with the field-induced selection of a ground state from a degenerate manifold of states in a frustrated magnet [16–19] while another is governed by the peculiar physics of metamagnetic transitions [20], by simply changing the direction of the applied field in the sample.

## 2 Experimental details

Single crystal samples of $SrEr_2O_4$, grown by the floating zone technique using an infrared image furnace as previously reported [21], were studied on the WISH neutron diffractometer [22] at the ISIS facility at the Rutherford Appleton Laboratory (United Kingdom) [23,24] as well as on the diffractometers D9 and D10 [25] at the Institut Laue-Langevin, Grenoble, France. On WISH, a dilution refrigerator inside a vertical-field cryomagnet provided a base temperature of 60 mK. The measurements were made in an applied field of up to 50 kOe. The sample was attached to an oxygen free copper holder with either the $a$ or $c$ axis vertical defining the horizontal $(0kl)$ or $(hk0)$ scattering planes. Although substantial coverage of the out-of-plane scattering is typically available on WISH due to a continuous array of position-sensitive detectors, the cryomagnet employed limited the vertical aperture. In order to maximise the diffuse signal, we opted for rather large, irregular shaped samples of about 250 mm$^3$ in volume.

For $H \parallel c$, the measurements were made in the temperature range from 60 mK to 0.8 K. Zero field measurements made at 15 K, i.e. well above $T_N$ were used to isolate the nuclear component of the scattering allowing for the extraction of magnetic intensity by direct subtraction. For integer $(hkl)$, only the $l = 0$ reflections could be reached, however, a broad diffuse scattering signal was observed in the $(hk-1/2)$ plane. For $H \parallel a$, only the $h = 0$ reflections could be reached for integer values $(hkl)$. For this direction of applied field, the experiment was performed prior to the addition of the second detector bank on the WISH instrument.

The data collected on WISH have been treated with the Mantid software [26] which takes into account vanadium normalisation as well as a correction for absorption. The diffuse scattering features and their evolution with an applied field are best seen through the two-dimensional intensity maps in reciprocal space, particularly after the higher-temperature background subtraction.

Measurements performed on the hot-neutron diffractometer D9 in zero field and four-circle geometry were used to derive the precise crystal structure of $SrEr_2O_4$ including the extinction parameters of the crystal used (see Appendix, Sec. A). A small cuboid sample of dimension $1.1 \times 3.7 \times 2.9$ mm$^3$ along the $a$, $b$ and $c$ axes, respectively, was employed for these measurements in order to reduce the beam absorption and problems due to extinction and multiple scattering. The wavelength of the monochromatic beam was fixed at 0.84 Å and with the sample temperature at 20 K, 541 unique reflections were collected.

D9 was then operated with a cryomagnet in normal-beam geometry for further measurements at 1.5 K in a field of 20 kOe applied along the $c$ axis in order to complement the WISH measurements. The wavelength was set at 0.51 Å and a set of 131 unique reflections in the accessible reciprocal space was measured. Each reflection was measured twice, with and without applied field in order to isolate the magnetic signal by direct subtraction.

An additional experiment was carried out using the D10 diffractometer using a dilution refrigerator and a vertical cryomagnet. The cuboid-shaped single-crystal sample of approximate dimensions $2.5 \times 1 \times 3$ mm$^3$ along the principal crystallographic axes was glued to the tip of a Cu pin with the $a$ axis vertical, i.e. parallel to the applied magnetic field. A wavelength of 2.36 Å was selected from a pyrolytic graphite monochromator and the sample was cooled to the base temperature of 50 mK. Data sets of 151 unique integer reflections were collected at 50 mK and at magnetic field values of 0, 6, 11 and 14 kOe, while a reduced data set of

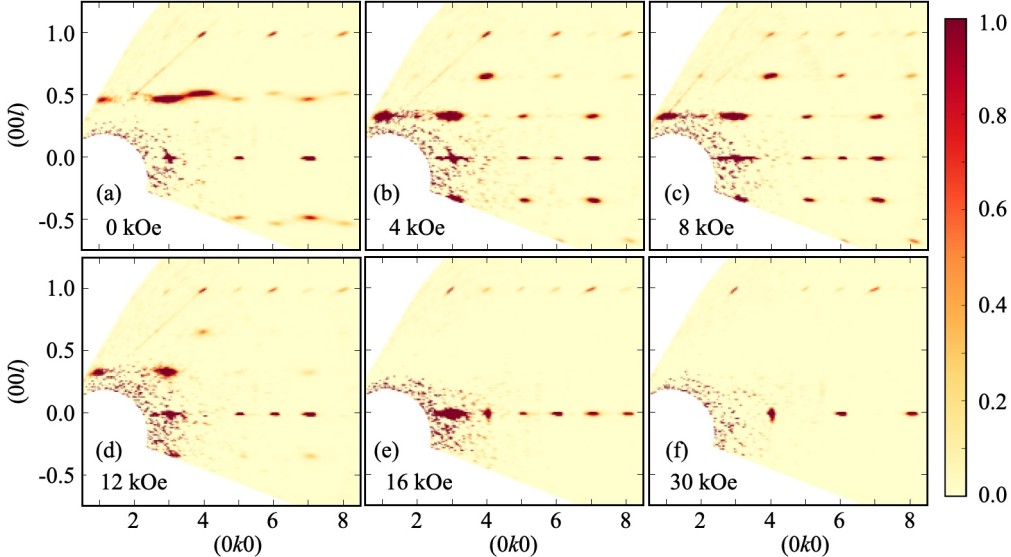

Figure 2: Single crystal neutron diffraction intensity maps of the $(0\,k\,l)$ plane measured in SrEr$_2$O$_4$ single crystal sample at $T = 60$ mK in different fields applied parallel to the $a$ axis. The high temperature background is subtracted to emphasise the magnetic contributions. We observe a clear field-evolution of the magnetic state of the Er2 ions from (a) zero-field collection of largely uncorrelated incommensurate spin chains through (b-d) the significantly more correlated UUD structures with $\mathbf{q} = (0\ 0\ \frac{1}{3})$ propagation vector and to (e,f) field-polarised phase characterised by the resolution limited $\mathbf{q} = 0$ peaks. The antiferromagnetic order formed by the Er1 ions is also seen as a set of $(0\,k\,l)$ peaks with integer $k$ and $l$ peaks present in all fields apart from (f) 30 kOe.

113 reflections was measured at 30 kOe. In the intermediate phase stabilised on the Er2 site 227 unique fractional reflections - modulated by the propagation vector $\mathbf{q} = (0\ 0\ \frac{1}{3})$ - were recorded at $T = 350$ mK and $H = 6$ kOe.

All data collected on D9 and D10 have been corrected for absorption using the Datap software [27] with a linear absorption coefficient of $\mu = 0.9270$ cm$^{-1}$ for SrEr$_2$O$_4$. The FULLPROF program [28] was used for magnetic and crystal structure refinements reported in this paper.

Very low temperature magnetisation measurements were performed with a purpose built SQUID magnetometer equipped with a dilution refrigerator [29] down to 90 mK on a 66 mg cuboid single crystal sample of dimension $2.2 \times 2.1 \times 2.1$ mm$^3$. The sample was glued with GE varnish on a copper sample holder to ensure a good thermalisation. The magnetic field was aligned along the $a$ axis, which was parallel to one of the short sides of the cuboid.

## 3 Experimental results

### 3.1 Field along the a axis

Figure 2 shows the evolution of the magnetic correlations in the $(0\,k\,l)$ scattering plane in SrEr$_2$O$_4$ with the field applied along the $a$ axis as measured on the WISH diffractometer. In zero field, Fig. 2a, we observe the coexistence of Bragg peaks associated with the Er1 antiferromagnetic ordering and the diffuse scattering corresponding to the development of the short-range quasi one-dimensional magnetic order on the Er2 sites. The zero-field diffuse scattering rods of intensity running along the $k$ direction for $l \approx \pm 1/2$ and other half-integer

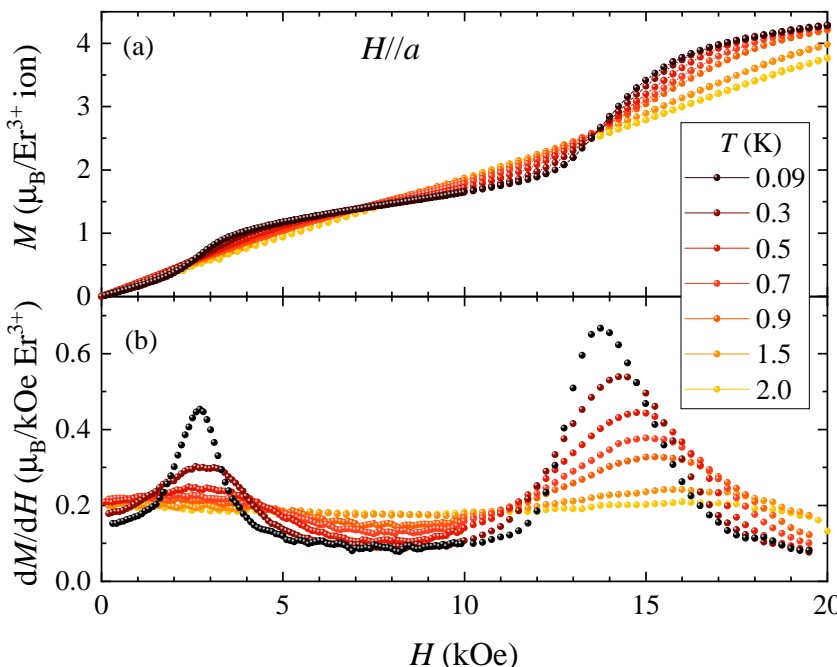

Figure 3: Field dependence of (a) magnetisation, $M(H)$, and (b) its derivative, $dM/dH$, measured in single crystal of the $SrEr_2O_4$ for $H \parallel a$ at different temperatures. The 1/3-magnetisation plateau is most pronounced at the lowest temperature, but it remains clearly visible at temperatures well above the zero-field ordering temperature, $T_N = 0.75$ K.

positions have previously been seen in single crystals neutron diffraction, using both polarised and unpolarised neutrons [15].

On application of a magnetic field of only 2 kOe, the rods of scattering intensity gain considerable modulation along the $k$ direction and effectively transform into a collection of diffuse features, which are much more defined and which start to deviate significantly from the half-integer positions in the $l$ direction. On increasing the magnetic field to 4 kOe, the diffuse peaks move to the positions $(0\,k\,^{\pm l}/_3)$ where $k$ is an integer and $l = 1, 2$ (see Fig. 2b). On further increase of the field to 8 (Fig. 2c) and 10 kOe (the full record of the measurements in various fields is given in Appendix, Sec. B) the diffuse peaks remain at the same positions, while their intensities change marginally. The strongest field at which the diffuse peaks are still visible (although with significantly reduced intensity) is 12 kOe (Fig. 2d).

It is rather instructive to compare the neutron diffraction results with the detailed magnetisation measurements. Fig. 3 summarises the field evolution of the magnetisation curve $M(H)$ measured for $H \parallel a$ at different temperatures, the lowest being 90 mK, which is similar to the diffraction measurement temperature. The comparison makes it obvious that the field range, 4 to 12 kOe, coincides with the appearance of the 1/3-magnetisation plateau, typically associated with UUD magnetic structures in triangular, *zigzag* and other frustrated and low-dimensional magnets [17,30]. From a neutron diffraction perspective, such a UUD structure can be decomposed into a sinusoidally modulated component with a propagation vector $\mathbf{q} = (0\,0\,^1/_3)$ and a ferromagnetic component. For equally sized moments, the ferromagnetic component needs to be exactly $^1/_4$ of the antiferromagnetic amplitude, although there is no general requirement for such a restriction and a UUD structure with variable size moments have been found, for example, in $BaDy_2O_4$ [31].

The ferromagnetic component of the UUD order can be derived by considering the field

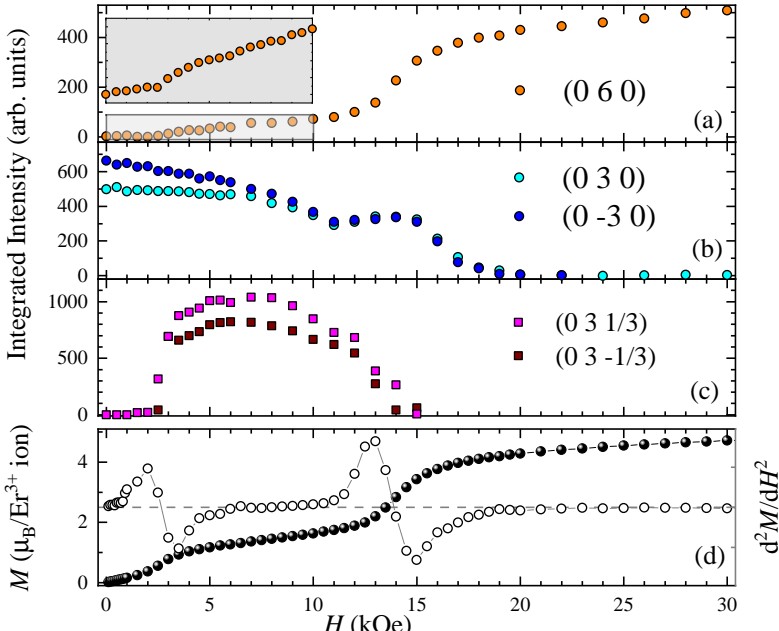

Figure 4: Integrated intensities recorded at $T = 50$ mK on the D10 diffractometer as a function of the magnetic field applied along the $a$ axis for (a) the (060) reflection representing the ferromagnetic component, (b) the $(0 \pm 3\, 0)$ reflections characteristic of the zero-field antiferromagnetic structure and (c) the $(0\, 3\, \pm 1/3)$ reflections representing the modulated spin component of the UUD structure. The inset focuses on the low-field region emphasizing the field-induced phase transition at about 2.5 kOe by showing the data taken with higher density of points and longer monitor time. (d) Magnetisation curve, $M(H)$, and its second derivative, $d^2M/dH^2$, measured at $T = 90$ mK.

evolution of reflections located at the $(0\, k\, l)$ positions with $k + l =$ even. In zero applied field, the (030), (050), (070), (041), and (061) reflections are observed as sharp resolution-limited Bragg peaks and originate from the Er1 antiferromagnetic structure previously determined [15]. On application of the field along the $a$ axis, these reflections are only weakly affected until the field value reaches approximately 10-12 kOe. Thus despite a clear rearrangement of the Er2 magnetic moments in an applied field, the low-temperature low-field Er1 magnetic structure remains stable. The reflections such as (040), (060) and (080) show an intensity increase appearing along with the stabilization of the Er2 UUD structure. Given the magnetisation results, Fig. 3, we expect these reflections to originate from a small ferromagnetic polarisation of the moments along the field.

Further increase of the magnetic field applied along the $a$ axis (see Fig. 2f) pushes SrEr$_2$O$_4$ into a polarised state, where the magnetic moments on both sites are pointing along the field. This state is characterised by the presence of only sharp magnetic Bragg peaks at the $(0\, k\, l)$ positions with integer values of $k$ and $l$ and $k + l =$ even. It becomes apparent that the Er1 moments undergo a reorientation, since the (030), (050), (070), (041), (061) and (081) reflections – all originating from the Er1 antiferromagnetic order – are absent at high fields.

A detailed field dependence of the intensity of magnetic peaks was obtained on the D10 diffractometer by integrating rocking scans of the Bragg reflections in fine field steps at $T = 50$ mK. The field evolution of the modulated component of the UUD structure, the $(0\, 3\, \pm 1/3)$ satellites, is compared to the ferromagnetic component, the (060) reflection, and to the zero-field magnetic structure, $(0 \pm 3\, 0)$ reflections, in Fig. 4. In agreement with the known zero-field

Table 1: Refined magnetic moment components $M$ of the Er1 and Er2 sites as a function of applied magnetic field, $H \parallel a$. Note that the $M_a$ only refers to the ferromagnetic component derived from the $\mathbf{q} = 0$ reflections. The $M_c$ component represents the amplitude of the long-range antiferromagnetic structure in zero field, in which only the Er1 site is involved.

| $H$, kOe | $M_a$ (Er1) | $M_a$ (Er2) | $M_c$ (Er1) |
|---|---|---|---|
| 0 | 0 | 0 | 3.9(3) |
| 6 | 0.6(4) | 0.8(4) | 3.5(1) |
| 11 | 1.4(1) | 2.4(1) | 3.4(1) |
| 14 | 1.9(2) | 3.2(2) | 3.8(2) |
| 30 | 3.3(2) | 7.3(5) | 0 |

magnetic structure the ferromagnetic (060) reflection, Fig. 4a, and the commensurate satellites, Fig. 4c, are extinct in the absence of an applied magnetic field and strong intensities are observed on the antiferromagnetic (0 ±3 0) reflections, Fig. 4b. At approximately 2.5 kOe a kink in the field dependence of the (060) reflection can be seen marking the onset of the transition into the UUD state, as shown the inset in Fig. 4a. It can be seen that the field in which there is a continuous increase of the intensity of the (060) reflection, panel Fig. 4a, coincides with the stability range of the magnetic satellites, panel Fig. 4c, while the intensities of the (0 ±3 0) reflections – representing the zero-field antiferromagnetic order – decrease. At approximately 14 kOe the magnetic satellites disappear while the (060) reflections starts to show a steep increase marking the transition into the field-polarised state. At roughly the same field strength the intensity of the (0 ±3 0) presents a weakly pronounced local maximum before it starts to decrease at higher fields, which suggests that both Er sites are aligned with the external magnetic field. In Fig. 4d we show the magnetisation data $M(H)$ collected at the lowest experimentally attainable temperature as well as its second derivative $d^2M(H)/dH^2$ in the same field range as the diffraction data to make the comparison between the techniques and to help identifying transition regions. We return to the comparison in section 4.

The process of field polarisation of magnetic moments was investigated by analyzing the integrated intensity data sets of $\mathbf{q} = 0$ reflections at various values of applied field and refining the magnetic structure models. Both Er sites have been included in the refinement, i.e. a non-zero spin component along the $a$ axis on the Er2 site represents the ferromagnetic component of the UUD structure (note that we deal with the modulated part below), while a finite spin component along $a$ for the Er1 site suggests a spin canting along the applied field. Table 1 lists the results of the refinement which show a continuous increase of the induced magnetic moment component of both Er sites with increasing magnetic field. At the same time the zero-field order of the Er1 site is hardly affected below 14 kOe as evidenced by the refined $c$ component. The local maximum seen in Fig. 4(b) is corroborated by a slight increase of the refined $c$ component in comparison to the lower field values.

In order to deduce the magnetic structure corresponding to the $\mathbf{q} = (0\,0\,1/3)$ satellites representation analysis was used to provide the magnetic symmetries compatible with the space group, the propagation vector and the involved Wyckoff sites. From the four possible irreducible representations shown in Table 2, $\Gamma_2$ provided the best agreement with the experimental data. Only the spin component $u$ along the $a$ axis was refined yielding an amplitude of 2.44(4) $\mu_B$ (in a field of 6 kOe).

Combining the incommensurately modulated and the $\mathbf{q} = 0$ component at 6 kOe, the resulting magnetic structure, viewed along the chains, consists of a succession of two parallel moments of 1.6(4)$\mu_B$ followed by an antiparallel moment of 2.0(4)$\mu_B$. Within the triangular ladders, the UUD magnetic chains are paired by a symmetry-imposed $\pi/2$ phase shift in the

Table 2: Basis vectors $\psi_n$ ($n = 1$-4) of the irreducible representations $\Gamma_n$ for the Er ions $m = 1$-4 at given fractional coordinates $(x, y, z)$ associated with a propagation vector $\mathbf{q} = (0\,0\,1/3)$. The components $u$, $v$ and $w$ connected to the spin $S_{\Gamma_n}^m$ have been refined according to their constraints (an overline indicates a negative number). The phase factor $\phi = \exp(-2\pi i\mathbf{qr})$ results from the translation $\mathbf{r} = (0\,0\,1/2)$ associated with the screw axis parallel to the $c$ axis or $\mathbf{r} = (0\,1/2\,1/2)$ connected to the glide plane perpendicular to the $a$ axis.

| Atom $m$ | Position | $\psi_1$ | $\psi_2$ | $\psi_3$ | $\psi_4$ |
|---|---|---|---|---|---|
| 1 | $\begin{pmatrix} x \\ y \\ 1/4 \end{pmatrix}$ | $\begin{pmatrix} u \\ v \\ w \end{pmatrix}$ | $\begin{pmatrix} u \\ v \\ w \end{pmatrix}$ | $\begin{pmatrix} u \\ v \\ w \end{pmatrix}$ | $\begin{pmatrix} u \\ v \\ w \end{pmatrix}$ |
| 2 | $\begin{pmatrix} \bar{x} \\ \bar{y} \\ 3/4 \end{pmatrix}$ | $\phi \cdot \begin{pmatrix} \bar{u} \\ \bar{v} \\ w \end{pmatrix}$ | $\phi \cdot \begin{pmatrix} \bar{u} \\ \bar{v} \\ w \end{pmatrix}$ | $\phi \cdot \begin{pmatrix} u \\ v \\ \bar{w} \end{pmatrix}$ | $\phi \cdot \begin{pmatrix} u \\ v \\ \bar{w} \end{pmatrix}$ |
| 3 | $\begin{pmatrix} \bar{x} + 1/2 \\ y + 1/2 \\ 3/4 \end{pmatrix}$ | $\phi \cdot \begin{pmatrix} u \\ \bar{v} \\ \bar{w} \end{pmatrix}$ | $\phi \cdot \begin{pmatrix} \bar{u} \\ v \\ w \end{pmatrix}$ | $\phi \cdot \begin{pmatrix} u \\ \bar{v} \\ \bar{w} \end{pmatrix}$ | $\phi \cdot \begin{pmatrix} \bar{u} \\ v \\ w \end{pmatrix}$ |
| 4 | $\begin{pmatrix} x + 1/2 \\ \bar{y} + 1/2 \\ 1/4 \end{pmatrix}$ | $\begin{pmatrix} \bar{u} \\ v \\ \bar{w} \end{pmatrix}$ | $\begin{pmatrix} u \\ \bar{v} \\ w \end{pmatrix}$ | $\begin{pmatrix} u \\ \bar{v} \\ w \end{pmatrix}$ | $\begin{pmatrix} \bar{u} \\ v \\ \bar{w} \end{pmatrix}$ |

magnetic moment sequence, see Fig. 1c. We believe that this structure is largely responsible for the appearance of a one-third magnetisation plateau shown in Fig. 3.

## 3.2 Field along the c axis

Figure 5 captures the main result of the experiment with $H \parallel c$ on the WISH diffractometer. In zero field [see Fig. 5(a)], the diffraction pattern in the $(h\,k\,0)$ scattering plane consists of sharp, resolution-limited Bragg peaks representing the $\mathbf{q} = 0$ magnetic order of the Er1 site in full agreement with the previous powder [14] and single crystal [15] neutron diffraction results. Our further single-crystal data collected on the same instrument under similar conditions, $H = 0\,\text{kOe}$, $T \approx 0.6\,\text{K}$, (not shown) confirm the previously established structure deduced from powder neutron diffraction [14]: a parallel alignment of the magnetic moments on Er1 sites forming the chains running along the $c$ axis with nearest-neighbour chains paired antiferromagnetically and only a short-range antiferromagnetic order on the Er2 sites with the moments along the $a$ axis.

Application of a moderate field of 3 kOe (see Fig. 5b) causes significant decrease of the intensity and almost the disappearance of the sharp Bragg peaks, which are replaced by broad diffuse scattering features. The broad peaks are centred around the same $\mathbf{q} = 0$ positions as the sharp peaks suggesting that the correlation length of the antiferromagnetic state becomes limited. The region of fields where diffuse scattering dominates the pattern, 2 to 5 kOe, coincides with an abrupt increase of magnetisation during a metamagnetic transition [32]. The shape of a peculiar lozenge pattern formed by the diffuse scattering features in these fields is practically identical to the high-temperature ($T > T_N = 0.75$ K) signal seen in zero-field

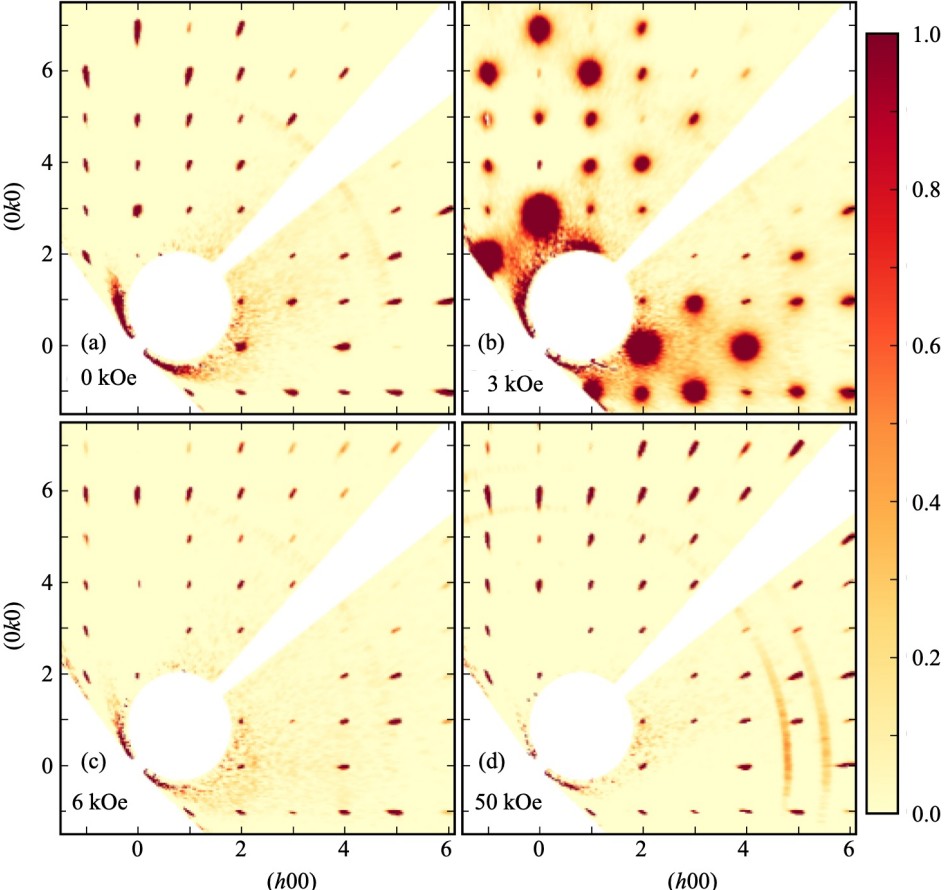

Figure 5: Single crystal neutron diffraction intensity maps of the $(h\ k\ 0)$ plane measured in $SrEr_2O_4$ at $T = 60$ mK in different magnetic fields applied parallel to the $c$ axis. The high temperature background has been subtracted to emphasise the magnetic contributions. The evolution followed is from (a) zero-field antiferromagnetic order of the Er1 moments through (b) short-range state during the metamagnetic transition to (c,d) field-polarised state which involves both Er1 and Er2 sites.

measurements and also found in the Monte Carlo simulations when the Er1 and Er2 ladders are weakly coupled [32].

Figure 6 compares the intensity and shape of several magnetic reflections measured at $T = 60$ mK for different field values. Two Gaussian functions were used to fit the asymmetric diffuse components of the magnetic scattering while the asymmetric back-to-back exponential function was employed for the sharp components. For the strong magnetic reflections associated with the zero-field magnetic phase (peaks such as (030), (160), and (400)) their diffuse component reaches maximum intensity for $H$ around 2-3 kOe while in this field they still contain a sharp, resolution-limited component. Since the intensity of the (060) reflection is zero in the same field, it is clear that this remaining sharp component does not originate from the field-polarised phase stabilised above 6 kOe, but rather from the original long-ranged antiferromagnetic order being unaffected throughout the transition.

The insets to Fig. 6 detail the field dependence of the intensity and the width of the diffuse scattering component, providing a direct measurement of a magnetic correlation length in the $a$-$b$ plane of $SrEr_2O_4$. The correlation length is field and temperature dependent, varying approximately from 100 to 600 Å.

Increasing the field up to 6 kOe is sufficient to restore a long-range order, (see Fig. 5c) as

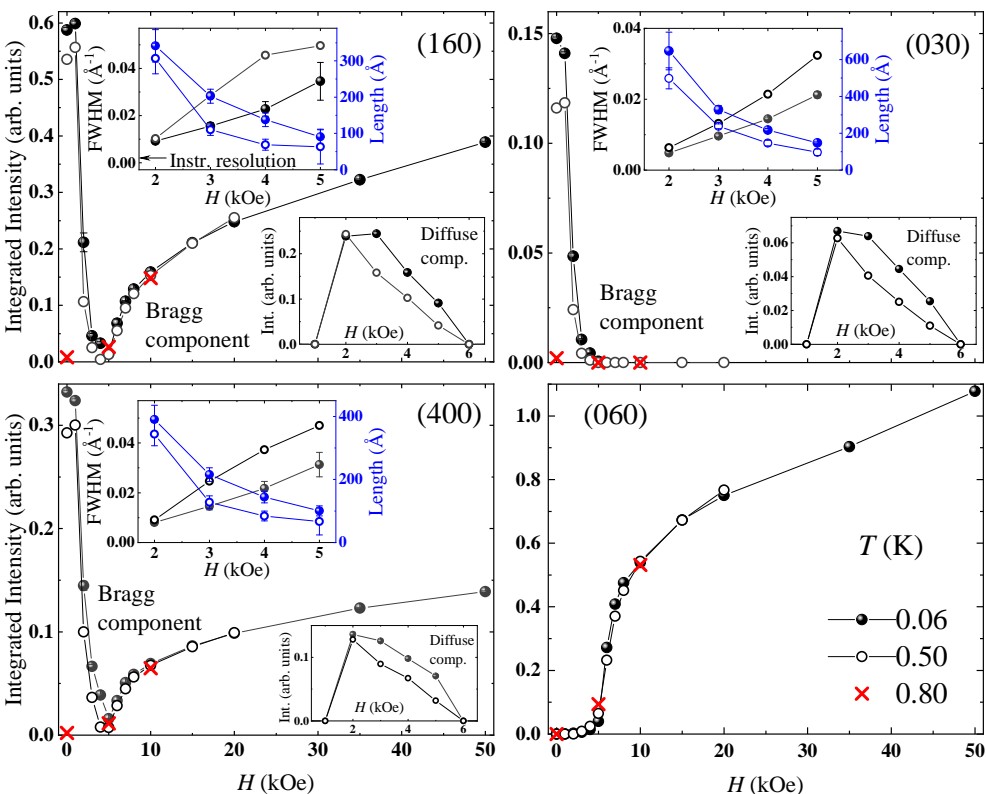

Figure 6: Main panels – field dependence of the integrated intensity of the sharp resolution-limited component of the four magnetic peaks, (160), (030), (400) and (060), for $H \parallel c$. The lower insets show the intensity of the broad magnetic diffuse scattering components as a function on an applied field, while the upper insets show the field variation of the width of these components and the associated correlation length. While the integrated intensity is shown in arbitrary units, the same manipulation procedures have been applied to all four reflections, making them directly comparable.

strong resolution-limited Bragg peaks appear again, but their intensity is different from those in the zero-field because the antiferromagnetic order is replaced by a field-polarised state. This is a stable state, not particularly sensitive to temperature increases. For example in a field of 20 kOe, the intensity of Bragg peaks remains practically the same on warming up to 0.8 K, well above the $T_N$ in zero field.

It is important to emphasize the unusual history dependence demonstrated by $SrEr_2O_4$ for $H \parallel c$ as the system remains in the short-range state when the field is ramped back to zero from the higher values. At low temperatures, after returning to zero field, the magnetic intensity maps show the presence of diffuse scattering superimposed with the sharp Bragg peaks (see Fig. A5 in Appendix, Sec. B). In order to return to a proper zero-field state, the sample needs to be warmed above the $T_N = 0.75$ K and cooled again in a zero field.

The WISH diffractometer has the substantial coverage of the out-of-plane scattering which has allowed us to access the diffuse scattering features associated with the Er2 sites and follow their field dependence for $H \parallel c$. Although the scattering was limited to rather large wave vectors ($k$ in the range 8 to 12 for the features at the $(hk\frac{1}{2})$ positions) and therefore was low in intensity, it still made it possible to reach important conclusions. The results imply that the

structure formed by the Er2 moments is not very sensitive to the lower values of the external field applied along the $c$ axis, the intensity drops rapidly above the metamagnetic transition, it still remains present in 10 kOe and completely disappears in 20 kOe (see Fig. A6 in Appendix, Sec. B). This observation is consistent with a picture of a gradual polarisation of the Er2 sites by the applied transverse field.

In order to extract the magnetic moments on both Er sites in the field polarised state we analysed the monochromatic data set from the D9 diffractometer collected at $T = 1.5$ K and $H = 20$ kOe. Each reflection was measured twice, with and without applied magnetic field to allow for the isolation of the magnetic component of the scattering. The refinement returns the values of 5.4(1) and 1.9(1) $\mu_B$ for the moments on Er1 and Er2 sites, respectively, if a ferromagnetic model is presumed on both Er sites. The average magnetic moment is then 3.65 $\mu_B$ per Er ion for this value of a field applied along the $c$ axis in a good agreement with the previous magnetisation measurements [14, 32].

## 4  Discussions

The experimental findings for $SrEr_2O_4$ in magnetic field applied along the $c$ axis draw immediate comparisons with the properties of metamagnets, the antiferromagnets which in applied field undergo a phase transition into a state with a relatively large magnetic moment and which have been extensively investigated in the past, both experimentally and theoretically [20, 33]. In metamagnets, quite often the transition from an antiferromagnetic ground state to a field-polarised state (labeled as paramagnetic) is through an intermediate so called *mixed* phase in which domains of the antiferromagnetic and paramagnetic phases coexist. The origin of the mixed phase is the presence of the demagnetizing field which forces a first-order transition to be spread out over a range of applied fields [34]. Neutron diffraction measurements (see, for example, [35–37]) played a pivotal role in establishing the detailed magnetic phase diagrams of metamagnetic materials, including discontinuous transitions and tricritical points, however, we could not find in published literature any neutron diffraction measurements of the magnetic correlation length in a mixed phase (similar to those shown in Fig. 6). Perhaps this is not entirely surprising, as most of the early neutron scattering experiments were performed several decades ago, when the neutron instruments were lacking in both resolution and reciprocal space coverage compared to the state-of-the-art diffractometers.

One feature that clearly distinguishes $SrEr_2O_4$ from other metamagnets is the presence of the second sublattice which does not participate directly in the metamagnetic transition but still influences significantly the paramagnetic phase. In this respect the state near $T_N$ and in low fields is exposed to a weak staggered field from weakly correlated chains of magnetic moments on the Er2 sites, while the state near the saturation field at low temperatures is exposed to a significantly stronger field of the Er2 moments polarised along the field. It would be beneficial to explore in detail the magnetic $H - T$ phase diagram of $SrEr_2O_4$ using, for example, heat capacity or ultrasound velocity measurements which have been successfully used for mapping out phase transitions in similar compounds [2, 9].

Remarkably, the picture of broad diffuse-scattering peaks in the $(h\,k\,0)$ plane characterising the intermediate state of $SrEr_2O_4$ is identical to what is observed in $SrHo_2O_4$ in zero field [6]. Despite almost indistinguishable magnetisation curves for $H \parallel c$ in these two compounds [32], the sequence of the field induced phases is rather different. One could envisage two possible reasons for such a difference. The first one is to do with a possible splitting of the quasidoublet states of the $Ho^{3+}$ non-Kramers ions by crystalline electric fields (as briefly mentioned in Ref. [13]). Alternatively, in view of what we observed for $SrEr_2O_4$, it is possible that the mixed state in $SrHo_2O_4$ is so wide that it extends from practically zero field all the way to the

saturation field.

With the demagnetisation effects being significant, ideally one has to correct for them by subtracting the demagnetising field from the applied field. This procedure is straightforward for ellipsoidal samples for which the demagnetising factor depends only on sample shape, but is much more involved for non-ellipsoidal samples [38]. Faced with the necessity to use cuboid samples because of the low-temperature thermalisation requirements we report the uncorrected data. From previous magnetisation measurements [32] the demagnetising field was found always to be less than 5% of the applied field. It is also worth pointing out that our earlier measurements performed on the PRISMA instrument (technical details are given in Ref. [15]) using differently shaped samples returned results identical to those reported here.

The experimental results presented in the previous sections strongly suggest that a full description of the magnetic properties of $SrEr_2O_4$, would require a theoretical model which apart from the exchange and dipolar interactions includes finite magnetic anisotropy term(s) allowing for the deviations of the magnetic moment away from their easy-axis orientations in an applied magnetic field. However, it is still very useful to consider a simplified model consisting of two Ising-type sublattices on the two Er sites. Finding the general ground state solutions for such a model in an applied field is an interesting and complex problem in its own right [39], but one prediction for such a model seems to be especially relevant for $SrEr_2O_4$. The prediction is that for $H \parallel a$, the zero-field phase cannot directly transform into the UUD state and therefore an intermediate phase should exist. In this respect, an intriguing proposition is that the observed change in magnetisation rate in a transition regime between zero field and magnetisation plateau (see Fig. 3) marks the stability range of this additional phase. For lower fields (less than 1.2 kOe) the slope of the magnetisation curve is 0.16 $\mu_B$/kOe, on the plateau (between 5.5 and 10 kOe) it is 0.09 $\mu_B$/kOe while in a transition region it is 0.39 $\mu_B$/kOe. Within the Ising model [39], the predicted intermediate phase is characterised by the splitting of the Er1 site ladders into the two different types, one with the periodicity 2 along the $c$ axis (UDUD state) and another with periodicity 3 (UUD state). The ratio of the number of the UDUD to the UUD ladders is three to one, therefore the overall magnetisation is fixed at $\frac{1}{8}$ of the saturation magnetisation for the Er1 site. In the experiment, however, the magnetisation rapidly increases with field, perhaps a more realistic representation of the system is in fact a field-dependent ratio of the UDUD and UUD ladders. In a field of 2 kOe, neutron diffraction detects diffuse peaks at the positions $(0\,k\,\frac{1}{2}\pm\delta)$ with $\delta = 0.05$ (see Fig. A1 in Appendix, Sec. B). This observation is also consistent with the idea of a gradual evolution from the zero-field state characterised by $\delta = 0.02$ to the UUD state with $\delta = \frac{1}{6}$ through a field-dependent mixture of the UDUD to the UUD ladders.

In terms of theoretical modelling, the one-dimensional ANNNI model was previously used to describe the magnetic properties of $SrHo_2O_4$ and $SrDy_2O_4$ [7,11]. The model captures several features (including the UUD state for longitudinal fields) and allows for the introduction of the effective exchange constants, however, being minimalistic in nature, it ignores the interactions between the ladders and therefore excludes the possibility of the complex intermediate phases discussed above. While the magnetisation curves look similar for the magnetic field applied along the easy-axis in $SrHo_2O_4$, $SrDy_2O_4$ and $SrEr_2O_4$ [32], the long-range magnetic order in the RE1 sublattice is established only in $SrEr_2O_4$, making it a unique case.

In many respects the behaviour of $SrEr_2O_4$ is similar to what is found in spin ice materials, including the presence of magnetic disorder down to the lowest temperatures and the unusual field-induced transitions (such as three-dimensional Kasteleyn transition for $H \parallel [100]$ [40]). For $H \parallel [110]$ the spin ice breaks into magnetic chains [41] characterised by the appearance of metastable states and unusually slow dynamics, as well as non-ergodicity (in common with what is observed in $SrEr_2O_4$ for $H \parallel c$).

The analogy with the low-$T$ behaviour of spin ice materials might extend even further.

An interesting general analogy could be drawn here with the phenomenon of moment fragmentation [42–44] which gives rise to a coexistence of spin liquid and ordered states in other frustrated magnets. The $SrRE_2O_4$ systems often demonstrate a coexistence of ordered and disordered magnetic components (which reside on different RE sites) and if the concept of fragmentation is potentially extendable to them then the key question must be about quantum or classical origin of fragmentation.

## 5 Conclusion

The field evolution of the low-temperature magnetic structure of $SrEr_2O_4$ was studied by single-crystal neutron diffraction for the two directions of the applied magnetic field, $H \parallel a$ and $H \parallel c$ as well as by low-temperature magnetisation measurements for $H \parallel a$. Reflecting the highly anisotropic Ising-like nature of the compound which contains magnetic Er ions on two different crystallographic sites, Er1 and Er2, the field-induced transitions for $H \parallel a$ are mostly to do with the rearrangements of the magnetic moments on the Er2 sites, while for $H \parallel c$, the moments on the Er1 sites mainly determine the value and the properties of the metamagnetic transition. For $H \parallel a$, the applied field induces a transition to an extended, but still not fully long-range ordered UUD state of the Er2 sites, which remains stable in a significantly wide range of the fields, 4 to 12 kOe, before changing to the state with all Er2 moments parallel to the field. For $H \parallel c$, the applied field initially destroys the long-range antiferromagnetic order of the Er1 sites and induces a much shorter-range order during the metamagnetic transition at 2 kOe prior to establishing a long-range ferromagnetic-like state of the Er1 moments at 6 kOe.

The development of a detailed theoretical model is now required to fully appreciate the fascinating sequence of field-induced magnetic phases found in $SrEr_2O_4$.

## Acknowledgements

We are grateful to B.Z. Malkin and Yu.I. Dublenych for numerous discussions of the magnetic properties of the $SrRE_2O_4$ compounds, to T.J. Hayes for the help with sample preparations and initial neutron diffraction measurements as well as to S.T. Bramwell and M.R. Lees for careful reading the manuscript and helpful suggestions. We would also like to acknowledge the expertise and dedication of the low-temperature groups at both ISIS and Institut Laue-Langevin. The work at the University of Warwick was supported by EPSRC through grants EP/I007210/1, EP/M028771/1, and EP/T005963/1.

## A Nuclear structure refinement

Table AI shows the refined parameters of the nuclear structure of $SrEr_2O_4$ obtained from the D9 single crystal neutron diffraction data collected at 20 K in zero field using a four-circle geometry. The high quality data set allowed for a high precision in the refinement of the atomic positions ($R_{Bragg} = 3.57$). The nuclear structure parameters are used in the magnetic structure refinements.

Table AI: Refined ionic positions of $SrEr_2O_4$ obtained from neutron diffraction data measured at 20 K. The dimensions of the orthorhombic ($Pnam$) unit cell are $a = 10.0189$ Å, $b = 11.8611(9)$ Å and $c = 3.3893$ Å. All the ions occupy the $4c$ Wyckoff sites with general positions $(x, y, 0.25)$.

| Atom | $x$ | $y$ |
|------|-----|-----|
| Sr | 0.7523(1) | 0.6500(3) |
| Er1 | 0.4225(2) | 0.11 |
| Er2 | 0.4232(8) | 0.6120(9) |
| O1 | 0.2119(5) | 0.1745(9) |
| O2 | 0.1262(3) | 0.4799(6) |
| O3 | 0.5146(2) | 0.7836(1) |
| O4 | 0.4250(1) | 0.4222(3) |

# B   Neutron Diffraction Intensity maps

Intensity maps have been collected on the WISH time-of-flight diffractometer at ISIS for a single sample orientation with $H \parallel a$ and for the two different sample positions with $H \parallel c$. In all cases the high-temperature background was subtracted to emphasize the magnetic contributions.

### B.0.1   Scattering plane $(0kl)$, $H \parallel a$

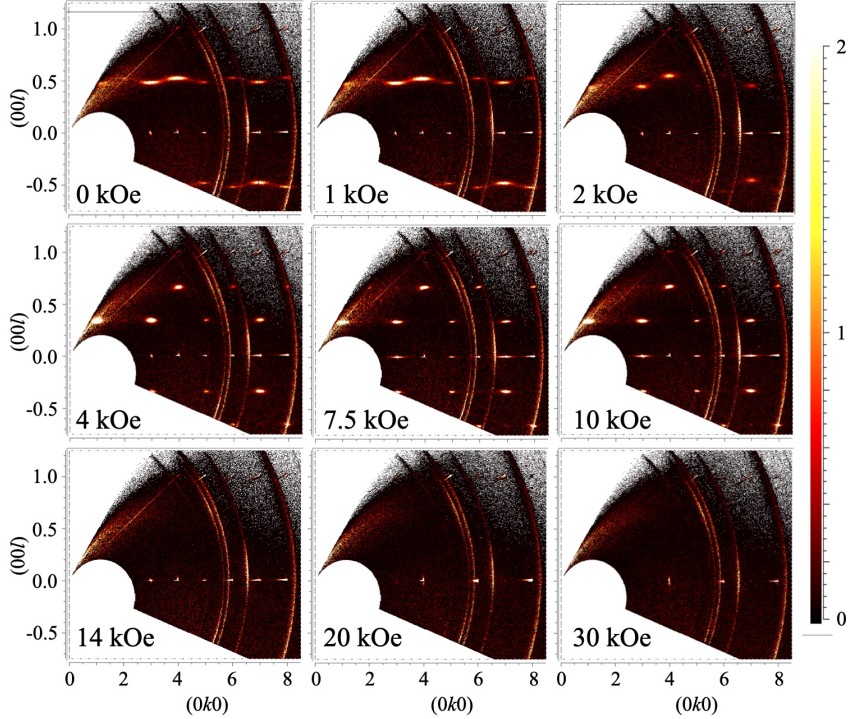

Figure A1: Neutron diffraction intensity maps (background subtracted) of the $(0kl)$ plane measured on the WISH instrument at $T = 60$ mK in different fields applied parallel to the $a$ axis.

Fig. A1 gives the detailed record of the field dependence of the magnetic diffraction intensity in the $(0kl)$ scattering plane for the field applied along the $a$ axis. No significant history

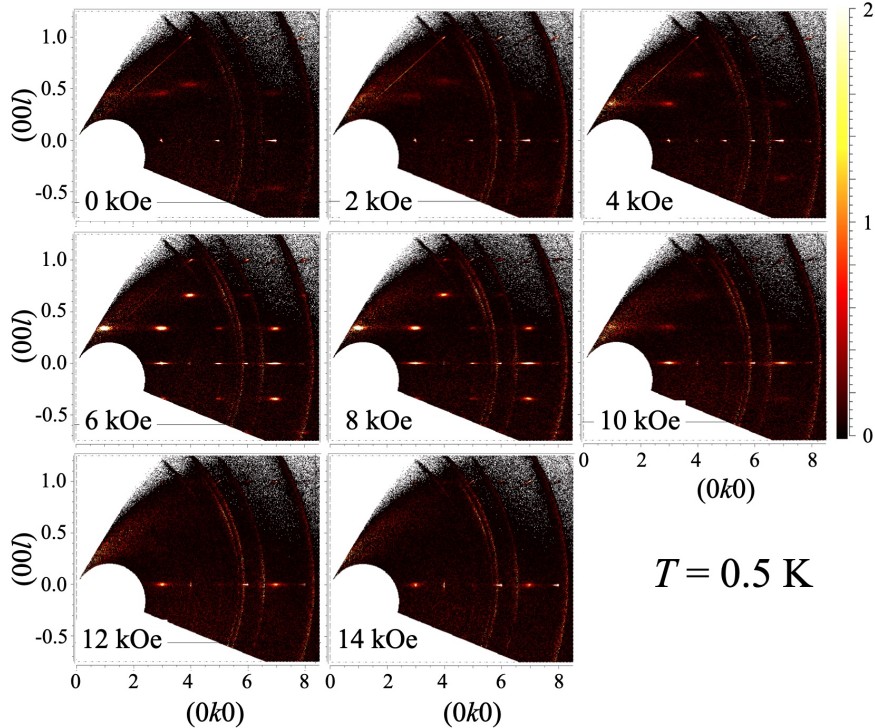

Figure A2: Neutron diffraction intensity maps (background subtracted) of the $(0kl)$ plane measured on the WISH instrument at $T = 0.5$ K in different fields applied parallel to the $a$ axis.

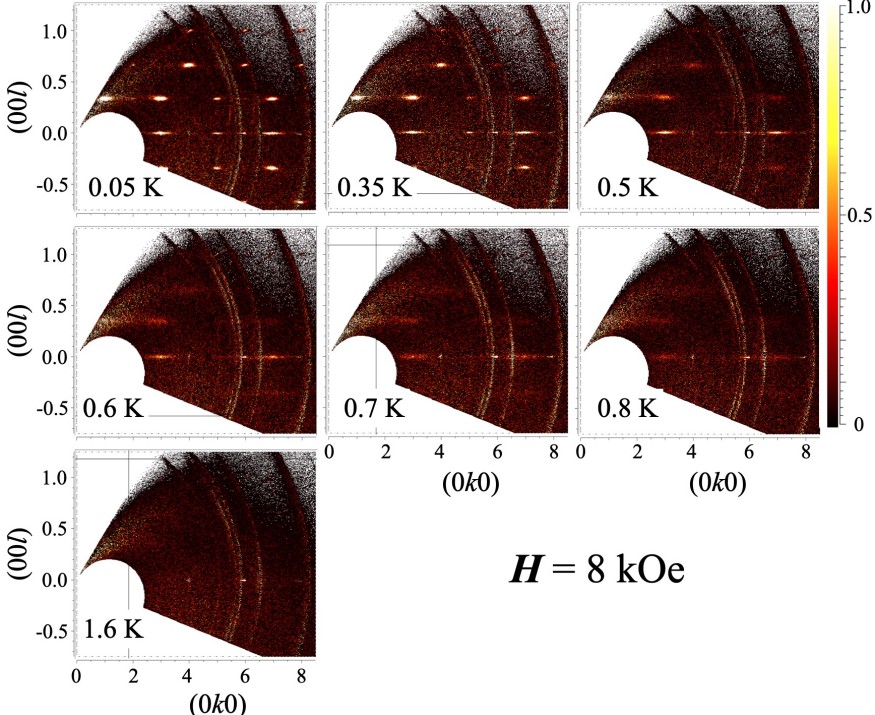

Figure A3: Neutron diffraction intensity maps (background subtracted) of the $(0kl)$ plane measured on the WISH instrument at different temperatures for a field of 8 kOe applied along the $a$ axis.

dependence has been found during the measurements, the results were largely reproducible for increasing and decreasing fields for several runs.

Fig. A2 shows the field dependence taken at $T = 0.5$ K. Compared to the base temperature (60 mK), an overall decrease in intensity is visible for all fields.

Fig. A3 follows the temperature dependence of the magnetic diffraction intensity in the $(0kl)$ scattering plane for the field $H = 8$ kOe applied along the $a$ axis. The measurements indicate a gradual reduction of the intensity of the diffuse features at the positions $(0\,k\,{}^{\pm l}\!/3)$ corresponding to the UUD magnetic structure.

**B.0.2 Scattering plane $(hk0)$, $H \parallel c$**

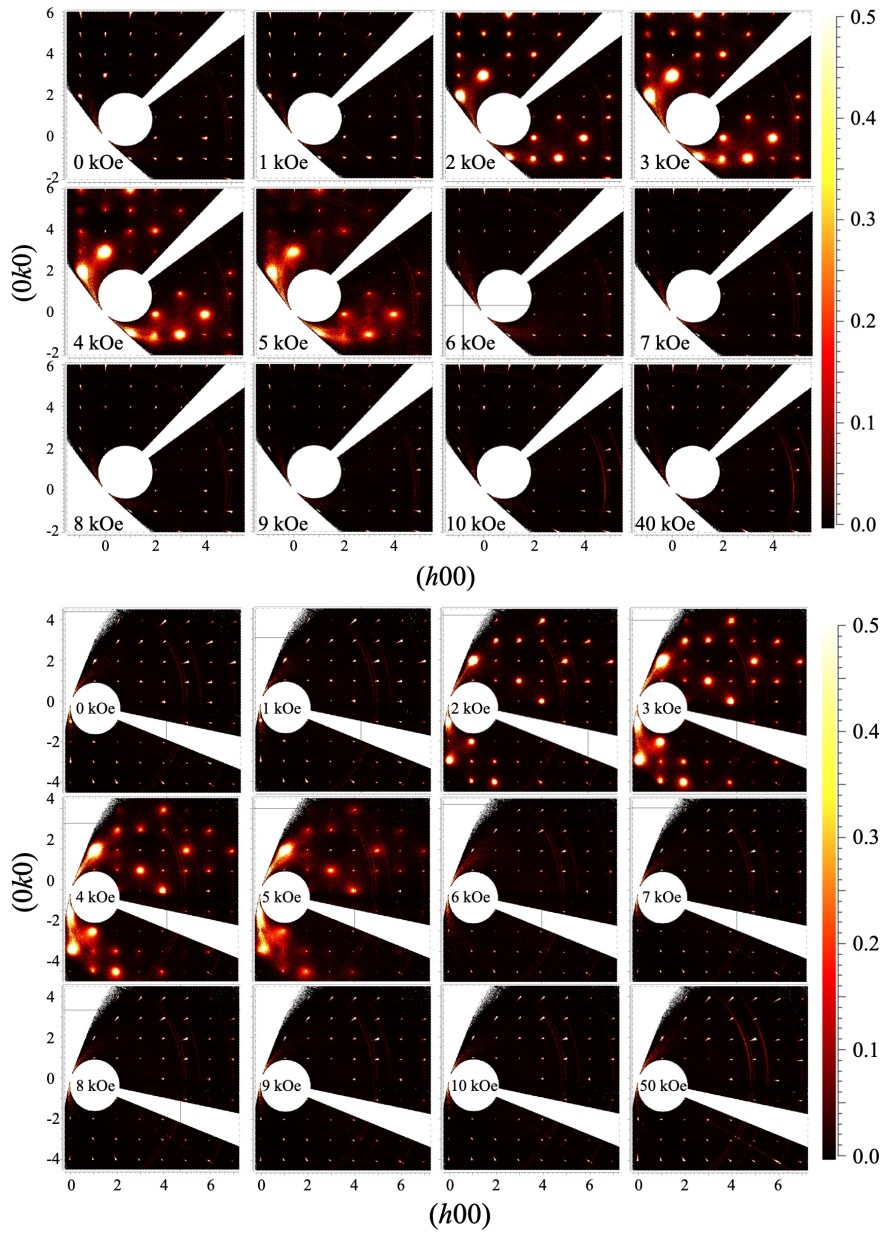

Figure A4: Neutron diffraction intensity maps (background subtracted) of the $(hk0)$ plane for SrEr$_2$O$_4$ measured on the WISH instrument at $T = 60$ mK in different fields applied parallel to the $c$ axis.

Fig. A4 shows the field dependence of the magnetic diffraction intensity in the $(hk0)$ scat-

tering plane for the field applied along the *c* axis. Two sets of the intensity maps shown were collected for the two sample positions (different by a rotation around the vertical *c* axis of 27 degrees), allowing for access to different sections of the reciprocal space.

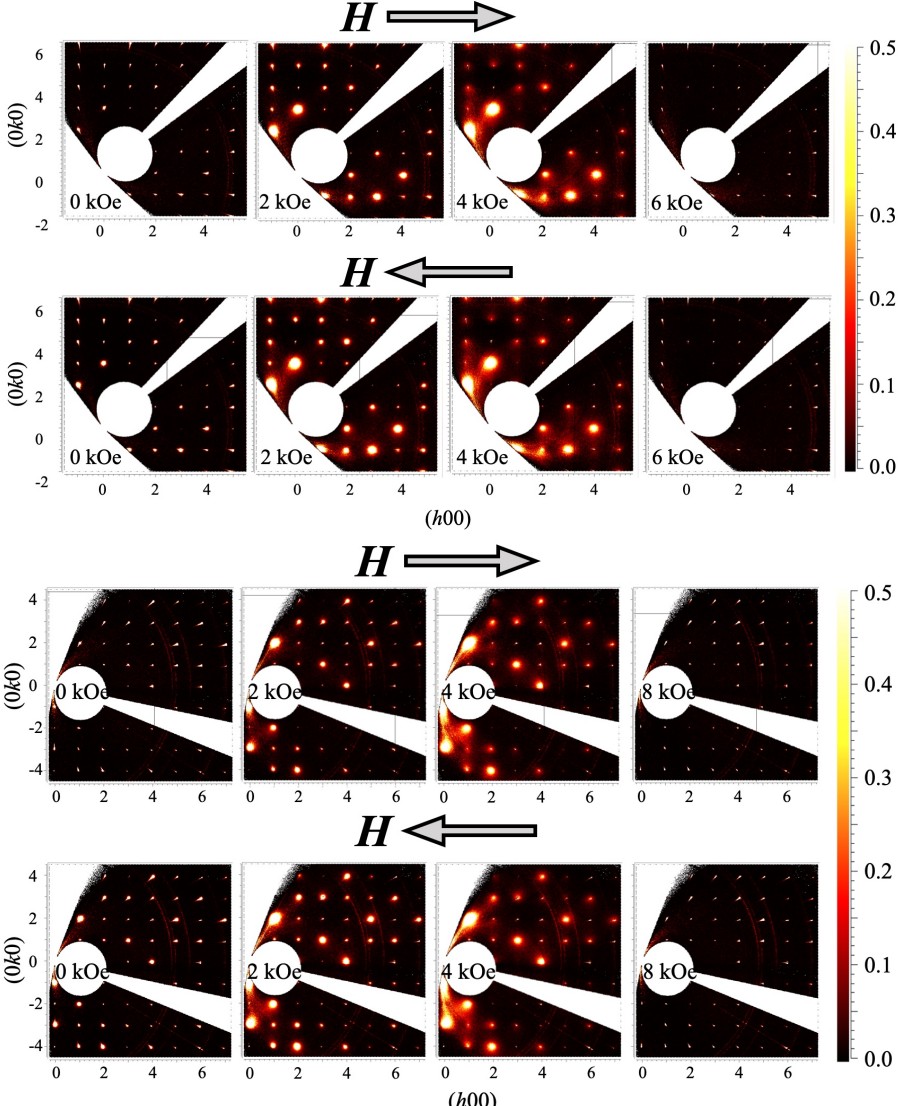

Figure A5: Comparison of the neutron diffraction intensity maps (background sub-tracted) in the $(hk0)$ plane measured in a field applied parallel to the $c$ axis. The temperature is 180 mK for the top panel and 60 mK for the bottom panel. For the bottom row in each panel, the field is decreasing from the maximum value of 30 kOe down to zero.

Fig. A5 emphasizes the observed field history dependence for $H \parallel c$. The intensity and the width of the magnetic peaks are different for increasing and decreasing fields. The zero-field maps contain a significant diffuse scattering components after exposure of the sample to a field of 30 kOe.

Fig. A6 shows the out-of-plane component of the magnetic scattering for $H \parallel c$ measured at $T = 60$ mK. The diffuse intensity seen at the $(h\, k\, \frac{\bar{1}}{2})$ positions gradually disappears with the applied field, it is not visible for the fields above 10 kOe. The maximum field applied to the sample was 50 kOe.

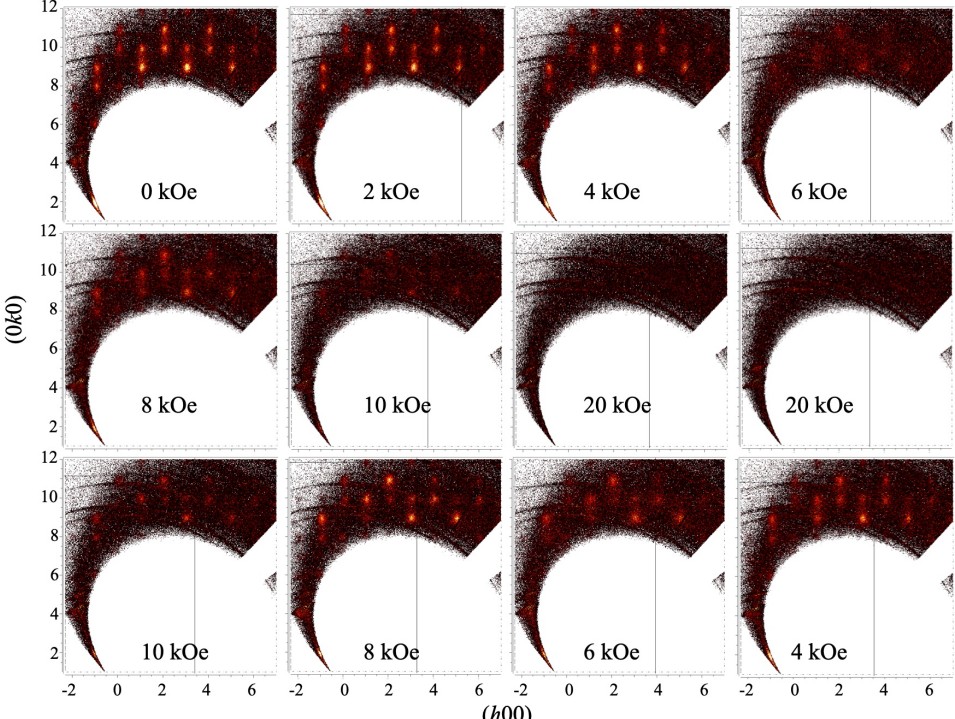

Figure A6: Out-of-plane scattering for different fields applied along to the $c$ axis at $T = 60$ mK. Top panel illustrates the disappearance of the $(h\,k\,\frac{\bar{1}}{2})$ signal in fields above 10 kOe and its restoration when the field is reduced.

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
