# Peer review of "Field-induced magnetic states in geometrically frustrated SrEr2O4"

_SciPost Physics, doi:SciPost Phys. 11, 007 (2021)_

## Round 1 · Referee Report · Anonymous (Referee 1) · 2021-4-28

Strengths

solid and very detailed experiments
careful analysis

Weaknesses

presentation somewhat hard to follow
figures mostly too crowded, insets too small, color coding doesn't very well

Report

The authors present a detailed study of a frustrated magnet, SrEr2O4, by means of magnetization and neutron diffraction in magnetic fields. In this material, the two crystallographically inequivalent Er sites also behave differently with respect to magnetic order: while the Er(1) site exhibits long-range magnetic order below 0.75K, for the Er(2) only short-order signatures have been reported. In the present manuscript, the in-field response along certain crystallographic directions of this of mixed magnetic order is reported and discussed. Most importantly, this study reveals a very different field response of the two Er sites and for the different crystallographic directions.

Overall, this is a very sound experimental study, with a very careful analysis of the experimental data. The conclusions reveal yet another highly unusual response to - in this case - magnetic fields as result of magnetic frustration. Altogether, I therefore conclude that in principle the paper meets the criteria for publication in terms of novelty and scientific soundness. Only, the presentation of the work, in particular regarding the figures in the manuscript, needs improvement (see requested changes). Only after these changes the paper maybe accepted for publication.

Requested changes

Figures are mostly too small, with various panels crammed together in one figure (example: figure 2, figures in appendix), various insets too small to read the numbers (figure 6), data points almost not visible (red crosses in figure 6), color coding inappropriate (figure 3c: dark purple vs. black as difference between (0 3 1/3) and (0 3 -1/3), intensity maps being basically black with some color spots visible to the trained eye only (figures in appendix). It makes the paper very hard to read ... the authors should optimize the figure presentation to the effect that it is not necessary to zoom in to 300%, and correspondingly modify the text as necessary.

---

## Round 1 · Referee Report · Anonymous (Referee 2) · 2021-5-2

Strengths

Thorough inelastic neutron-scattering study of single crystals of SrEr$_2$O$_4$ in an external magnetic field.

Weaknesses

1- Some figures lack clarity. 2- Missing bridge to Ising model description.

Report

The manuscript reports an unusual in-field behavior of SrEr$_2$O$_4$. This geometrically frustrated magnet hosts two crystallographically inequivalent Er ions that are both located on triangular zigzag ladders forming a honeycomb network, but only one site orders magnetically at low temperatures in zero magnetic field. The authors follow the sequence of peculiar field-induced states with detailed single-crystal magnetization and inelastic neutron scattering experiments. Among other phases, for the field applied parallel to the $a$ axis, an up-up-down structure is established in an intermediate field range.

This appears to be solid and interesting experimental work that in principle should be suitable for publication in SciPost Physics. However, some revisions may be appropriate before acceptance.

Apart from a list of suggested smaller changes (see below), I feel that a more detailed discussion of the magnetic Er$^{3+}$ ions in the present compound and their crystal-field level scheme would be helpful to close the gap to the Ising model, in particular in the discussion of section 4. For example, the magnetization data in Fig. 4(d) demonstrates that these are clearly not spin-1/2 Ising spins, at least not with a free-electron $g \approx 2$. It might also be helpful to discuss the local anisotropy axes. For example, if the Er1 and Er2 sites have different anisotropy axes, a transverse field component at least on one site could never be avoided such that there would always be some smearing expected. Such remarks may also help to elucidate the last sentence of the abstract where the authors state that full polarization of both Er sites is not achieved even in fields of 50 kOe.

Another global comment is that 2D color maps are esthetically nice ways of presenting data, but it is difficult to extract quantitative information from them. Thus, Figs. 2, 5, and A1-A6 may be a case for following the recommendation of publication of the underlying data, see item 5 of "General acceptance criteria" at https://scipost.org/SciPostPhys/about, preferably both the raw data as well as the background-subtracted data.

Requested changes

In addition to the revisions suggested in the Report, I have a few minor requests for changes:
A- Appendixes should be cited as such in the text and not as "Sec"tions.
B- In the fourth paragraph of the section "2 Experimental details" something is missing between "along the $a$, $b$ and $c$" and "respectively".
C- Figure 2 has low quality, in particular fonts are fuzzy.
D- The axis labels of Figs. 2 and A1-A6 are strange; I consider the notation of Fig. 5 to designate the coordinates -in this case $(hk0)$- preferable.
E- The text used to label Fig. 6 is a bit small.
F- Figures A1-A6 lack clarity (e.g., the intensity scale is difficult to read).
G- "Ising" and "Kastelyn" are names and should thus start with a capital letter in the titles of Refs. [35,36] and [37], respectively.

---

## Round 2 · Referee Report · Anonymous · 2021-6-23

Report

In response to my first report, the authors have improved the presentation of their work. I find the paper now to meet the criteria of scipost in terms of novelty, scientific soundness and presentation. I therefore recommend publication of the manuscript.

---

## Round 2 · Referee Report · Anonymous · 2021-6-24

Report

First of all, apologies for the spurious "in" before "elastic neutron scattering" in my previous report.

The aesthetics of color plots is of course at the discretion of the authors, but it is actually non-trivial to read off quantitative results from such a presentation. I am therefore grateful to the authors for adding the links [23-25] to the source data (even if processing them may be yet another story). It might just be a good idea to insist, e.g., at the top of page 4 (beginning of section 2) that these "references" are links to the data.

The other minor concern remains the Ising-model description of SrEr$_2$O$_4$. If, indeed, the anisotropy axes of the Er1 and Er2 sites are orthogonal, then a magnetic field that is aligned such as to be longitudinal on the Er1 sites is necessarily transverse on the Er2 sites and vice versa. However, the model of Ref. [39] assumes exclusively longitudinal fields. I admit that this reservation with respect to the applicability of the model of Ref. [39] to SrEr$_2$O$_4$ in a magnetic field is really a comment on Ref. [39] that is not under discussion here. Nevertheless, the authors could be a bit more careful about this point in their related discussion on page 13 of the present manuscript.

Apart from these details, I have reread the manuscript, and it does read well. Thus, I consider it in principle suitable for publication in SciPost Physics.

Requested changes

1- Delete a spurious "d" at the end of "mixed state" in the middle of the second line of page 13.

---

## Round 2 · Author Response

Dear Editors,

Following the Editorial Recommendation by the Editor-in-charge of our manuscript “Field-induced magnetic states in geometrically frustrated SrEr2O4” to resubmit after a minor revision, we are very pleased to attach a revised version of the manuscript. We believe that the modifications made have improved the manuscript and that it is now ready to be published by the SciPost.

---

## Round 2 · List of Changes

The Referee states that:
The manuscript reports an unusual in-field behavior of SrEr2O4. This geometrically frustrated magnet hosts two crystallographically inequivalent Er ions that are both located on triangular zigzag ladders forming a honeycomb network, but only one site orders magnetically at low temperatures in zero magnetic field. The authors follow the sequence of peculiar field-induced states with detailed single-crystal magnetization and inelastic neutron scattering experiments.

Our comment: It is a mistake to label the experiments as inelastic scattering. All the reported experiments are elastic neutron scattering, also known as neutron diffraction measurements.

The Referee states that:
Among other phases, for the field applied parallel to the a axis, an up-up-down structure is established in an intermediate field range. This appears to be solid and interesting experimental work that in principle should be suitable for publication in SciPost Physics. However, some revisions may be appropriate before acceptance. Apart from a list of suggested smaller changes (see below), I feel that a more detailed discussion of the magnetic Er3+ ions in the present compound and their crystal-field level scheme would be helpful to close the gap to the Ising model, in particular in the discussion of section 4. For example, the magnetization data in Fig. 4(d) demonstrates that these are clearly not spin-1/2 Ising spins, at least not with a free-electron g≈2. It might also be helpful to discuss the local anisotropy axes. For example, if the Er1 and Er2 sites have different anisotropy axes, a transverse field component at least on one site could never be avoided such that there would always be some smearing expected. Such remarks may also help to elucidate the last sentence of the abstract where the authors state that full polarization of both Er sites is not achieved even in fields of 50 kOe.

Our comment: The referee has unfortunately missed the information given in the text, see end of page 2 & start of page 3: “In SrEr2O4 in particular, relatively small variations in the positions of six nearest-neighbour oxygen ions which form distorted octahedra around the Er ions on two sites results in the occurrence of a strong Ising-like anisotropy with the easy-magnetisation direction along the c axis at the Er1 sites and along the a axis at the Er2 sites [13].” A few lines below, it is stated that “In SrEr2O4, the magnetic moments on the Er1 sites prefer to point along the c axis [13] … The magnetic moments on the Er2 site prefer to point along the a axis [13].” Combined with the fact that crystal electric field effects were carefully reported and discussed in our previous publication, ref. [13], we trust the Referee will agree that there is no reason to amend the manuscript.

The Referee states that:
Another global comment is that 2D color maps are esthetically nice ways of presenting data, but it is difficult to extract quantitative information from them. Thus, Figs. 2, 5, and A1-A6 may be a case for following the recommendation of publication of the underlying data, see item 5 of "General acceptance criteria" at https://scipost.org/SciPostPhys/about, preferably both the raw data as well as the background-subtracted data.

Our comment: We respectfully disagree with the above suggestion. The decision to include the intensity maps into the manuscript is not motivated by aesthetics, they are essential for following the field evolution of the magnetic structure(s) of SrEr2O4. They provide detailed information on the position, width and intensity of the magnetic scattering features. Without such an information it would be impossible to follow the field-induced changes of the magnetic structure, particularly as it varies in nature from long-range to short-range and back. The intensity maps provided us with crucial guidance when performing various field/temperatures scans on the reactor-based diffractometers, and we believe they will also serve as a guidance for the readers when navigating through the complex development of the magnetic correlations in SrEr2O4 in an applied filed.
We are also puzzled by the suggestion of providing the raw data as well as the background-subtracted data. We are happy to provide a link to the raw data (they are now included as 3 extra references), however, without the conversion from a time-of-flight format to the “intensity” vs “reciprocal space position”, the data are not practically usable. We see no reason to show both “raw” and background-subtracted intensity maps as the magnetic intensity in the former is masked by the strong nuclear features originating from both the sample itself and the sample’s environment equipment. The “raw” intensity maps therefore, do not reveal any new information regarding the magnetic correlations in SrEr2O4.

The Referee requested the changes --
In addition to the revisions suggested in the Report, I have a few minor requests for changes:
A- Appendixes should be cited as such in the text and not as "Sec"tions.
Our comment: We have followed recommendation and inserted the word “Appendix” every time we cite it.

B- In the fourth paragraph of the section "2 Experimental details" something is missing between "along the a, b and c " and "respectively".
Our comment: The amended version of the sentence now reads “A small cuboid sample of dimension 1.1x3.7x2.9 mm3 along the a, b and c axes, respectively…”

C- Figure 2 has low quality, in particular fonts are fuzzy.
Our comment: All original figures are of high quality, there are no fuzzy fonts in them. The quality reduction has happened because the PDF file has exceeded the submission size limit and needed to be compressed. We have now prepared most of the large figures in a “JPG” format with only a marginal decrease in quality, but a significant reduction in file sizes. If that is not sufficient, we are happy to provide PDF format files with no loss of picture quality if we are allowed to submit a large file.

D- The axis labels of Figs. 2 and A1-A6 are strange; I consider the notation of Fig. 5 to designate the coordinates -in this case (hk0)- preferable.
Our comment: We have now changed all the axis labels to the (h00), (0k0), (00l) format. This format was used in numerous previous publications (refs. [6,8,10,12,15]).

E- The text used to label Fig. 6 is a bit small.
Our comment: We have increased the font size in labels in Fig. 6.

F- Figures A1-A6 lack clarity (e.g., the intensity scale is difficult to read).
Our comment: We have increased the intensity scales in figures A1-A6.

G- "Ising" and "Kastelyn" are names and should thus start with a capital letter in the titles of Refs. [35,36] and [37], respectively.
Our comment: We have made the requested changes.
* * *
* * *
The Referee states that:
The authors present a detailed study of a frustrated magnet, SrEr2O4, by means of magnetization and neutron diffraction in magnetic fields. In this material, the two crystallographically inequivalent Er sites also behave differently with respect to magnetic order: while the Er(1) site exhibits long-range magnetic order below 0.75K, for the Er(2) only short-order signatures have been reported. In the present manuscript, the in-field response along certain crystallographic directions of this of mixed magnetic order is reported and discussed. Most importantly, this study reveals a very different field response of the two Er sites and for the different crystallographic directions.
Overall, this is a very sound experimental study, with a very careful analysis of the experimental data. The conclusions reveal yet another highly unusual response to - in this case - magnetic fields as result of magnetic frustration. Altogether, I therefore conclude that in principle the paper meets the criteria for publication in terms of novelty and scientific soundness. Only, the presentation of the work, in particular regarding the figures in the manuscript, needs improvement (see requested changes). Only after these changes the paper maybe accepted for publication.

Our comment:
Following the recommendations from both Referees, we have improved presentation of the manuscript.

The Referee requested the changes --
Figures are mostly too small, with various panels crammed together in one figure (example: figure 2, figures in appendix),
Our comment: We have increased the sizes of figure 2 and the figures in the Appendix.

various insets too small to read the numbers (figure 6),
Our comment: We have increased the data point sizes in figure 6.

data points almost not visible (red crosses in figure 6),
Our comment: We have increased the size of red crosses in figure 6.

color coding inappropriate (figure 3c: dark purple vs. black as difference between (0 3 1/3) and (0 3 -1/3),
Our comment: There is no panel (c) in figure 3, it is therefore likely that the referee meant figure 4(c). We do not share the Referee’s concern about colour coding in figure 4(c). As the intensities of the (0 3 1/3) and (0 3 -1/3) peaks should be identical in theory and are very similar in experiment, the colours chosen for their representation were deliberately close. The same approach was used in figure 4(b) when reporting the intensity of the (0 3 0) and (0 -3 0) peaks. We believe that the choice of the colour scheme in various figures is the authors’ privilege, not the referees, as long as the message of the figure is transparent and clear. We are convinced that figure 4 transmits a clear message on the presence or absence of the reflections of various types (ferromagnetic peaks, antiferromagnetic integer peaks, non-integer peaks) in various applied fields.
intensity maps being basically black with some color spots visible to the trained eye only (figures in appendix).
In order not to overload the manuscript with numerous figures, we have included only two figures of the intensity maps into the main text. The rest of the intensity maps are included as an Appendix, but it is perfectly possible to follow the manuscript without the use of Appendix. No particular training beyond a standard condensed matter course is required to read the maps. They show the positions, intensity and shapes of the magnetic reflections in the reciprocal space. In that respect a “basically black” map provides very useful information on the absence of the magnetic signal. “colour spots” provide the most direct information on the magnetic structure, its periodicity and dimensionality.

It makes the paper very hard to read ... the authors should optimize the figure presentation to the effect that it is not necessary to zoom in to 300%, and correspondingly modify the text as necessary.
Our comment: As stated above, we have improved presentation of the manuscript and hope that the Referee will now find it acceptable.

---

## Editorial Decision

published